# Effect of Fattening Period Length on Intramuscular and Subcutaneous Fatty Acid Profiles in Iberian Pigs Finished in the *Montanera* Sustainable System

**Dolores Ayuso [1], Ana González [2],*, Francisco Peña [2], Francisco I. Hernández-García [1] and Mercedes Izquierdo [1]**

[1] Department of Animal Production, Centro de Investigaciones Científicas y Tecnológicas de Extremadura, Badajoz, 06187 Guadajira, Spain; lolayuso@gmail.com (D.A.); francisco.hernandez@juntaex.es (F.I.H.-G.); mercedes.izquierdo@juntaex.es (M.I.)

[2] Department of Animal Production, University of Córdoba, 14071 Córdoba, Spain; pa1peblf@uco.es

* Correspondence: agmartinez@uco.es

**Abstract:** Twenty-four extensively reared Iberian pigs were used to study the influence of fattening period length (30, 60 or 90 days) on the fatty acid profiles of intramuscular and subcutaneous fat and the relationships between both profiles. Regarding fatty acid (FA) percentage, PUFA was greater in backfat and MUFA was greater in intramuscular fat (IMF), regardless of fattening period length. The longer fattening period increased MUFA content in backfat (which had a more marked change in oleic acid) and decreased PUFA content in backfat and IMF, but it did not affect SFA content. Within the three-layer subcutaneous backfat, SFA content was greater in the inner layer, MUFA was greater in the outer layer and PUFA was greater in both of these layers. The few differences in FA composition between both adipose tissues suggest that the changes due to the feeding regime are slow and, therefore, although the length of the fattening phase was increased, the fatty acid profile did not change substantially. The strong relationship between the FA profiles of IMF and backfat might be used to predict one profile from the other one when the latter is more readily available for sampling or analytical reasons.

**Keywords:** free-range; backfat layers; intramuscular fat; Iberian pig; subcutaneous fat

## 1. Introduction

Consumers nowadays demand high quality products with a particular taste or specific health-related properties, in which the fat content and fatty acid (FA) profile are important factors. The Iberian pig breed is an autochthonous porcine breed raised traditionally in the southwest of the Iberian Peninsula linked to the "dehesa" [1]. Its products, especially those derived from animals fattened under extensive, free-range conditions, feeding on acorns and grass in a silvopastoral system known as "montanera" from early November to late February [1], comply with these requirements [2] and their high content of oleic acid makes it healthier than other animal products [3]. This animal production system contributes to the conservation of the "dehesa" ecosystem (sparsely forested Mediterranean grassland) and also its preservation through the maintenance of traditional Iberian systems for the production of high-quality meat products [4]. Moreover, prior to any sustainability assessment, the acknowledgment of the traditional livestock production with Iberian pigs is a key factor.

Several factors can influence the lipid composition of pork, but it has been proven that the breed and the type of diet ingested by the pig at the end of the fattening period are the main factors [5,6].

The FA composition of the Iberian pig adipose tissues is diet-dependent [3] and the contents of the four FAs (palmitic, stearic, oleic and linoleic acid) in a sample of subcutaneous fat from the animal's

rear lumbar region are often used by the industry to classify the pig carcasses, and recent research has shown that the FA profile can be a potential indicator to discriminate Iberian hams according to the type of final fattening [7]. A number of studies have looked into the FA profile in adipose tissues of the Iberian pig, e.g., in backfat [8–12], in intramuscular fat [13,14] or in both [15,16], but few of them correlated the profiles obtained in the two types of fat tissue. This information might be relevant since it is easier to sample backfat than intramuscular fat (IMF). In addition, few studies have assessed the influence of the duration of the fattening period under the "montanera" system on the fatty acid profile.

The main objectives of this experiment were to analyze the influence of the duration of the fattening period on the fatty acid profile of intramuscular and subcutaneous fat (total fat and outer, middle and inner layers) of purebred Iberian pigs reared under free-range conditions ("montanera") and the relationship between the fatty acid composition of both fat depots.

## 2. Materials and Methods

### 2.1. Animals

Twenty-four pure-breed Iberian barrows of the Valdesequera strain were used in the Valdesequera research farm (Badajoz, Extremadura, Spain) of CICYTEX (Figure 1). The animals were raised under standard commercial conditions up to the beginning of the fattening period ("montanera"). Then, the animals were randomly distributed into three groups of eight pigs each according to the length of the fattening period—30, 60 and 90 days ($M_{30}$, $M_{60}$ and $M_{90}$, respectively), and fed under free-range conditions from November to January. Their diet consisted of acorns dry matter (DM): 588.2 g/kg, crude protein: 57.8 g/kg of DM, crude fat: 75.7 g/kg of DM, crude fiber: 68.6 g/kg of DM) and pasture (dry matter: 160.9 g/kg, crude protein: 309.2 g/kg of DM, crude fat: 4.1 g/kg of DM, crude fiber: 163.2 g/kg of DM, according to existing literature [13]). Body weight (BW) and age at the start of the fattening period were 128.8 ± 1.0 kg and 350 ± 10 days. The final BW ranged from 157.6 kg to 225.6 kg (Table 1).

**Table 1.** Effect of duration of fattening period on body weights and carcass traits (least squares means).

| Traits | Duration of Fattening | | | RSD | *p*-Value [2] |
| --- | --- | --- | --- | --- | --- |
| | Groups [1] | | | | |
| | $M_{30}$ (N = 8) | $M_{60}$ (N = 8) | $M_{90}$ (N = 8) | | |
| Body weight at start, kg | 128.90 | 129.20 | 128.40 | 0.40 | n.s. |
| Body weight at slaughter, kg | 157.60 [a] | 190.40 [b] | 225.60 [c] | 28.40 | *** |
| Carcass weight, kg | 119.60 [a] | 163.00 [b] | 193.10 [c] | 30.80 | *** |
| Loin, kg | 2.10 | 2.20 | 2.50 | 0.16 | n.s. |
| Loin area, cm [2] | 27.10 | 28.8 | 31.4 | 1.80 | n.s. |
| Intramuscular fat, g/100 g | 2.70 [a] | 4.70 [ab] | 6.20 [b] | 1.50 | ** |
| Backfat thickness | | | | | |
| Total, cm | 5.60 [a] | 7.63 [b] | 9.56 [c] | 1.65 | *** |
| Outer layer, cm | 1.26 [a] | 1.59 [b] | 1.63 [b] | 0.17 | * |
| Middle layer, cm | 2.51 [a] | 3.62 [b] | 3.93 [b] | 0.62 | *** |
| Inner layer, cm | 1.84 [a] | 2.42 [a] | 4.00 [b] | 0.94 | *** |

[1] $M_{30}$ = Thirty days on fattening; $M_{60}$ = Sixty days on fattening; $M_{90}$ = Ninety days on fattening. [2] n.s.: not significant; * $p \leq 0.05$; ** $p \leq 0.01$; *** $p \leq 0.001$. [a,b,c] Values within a row with different superscript differ at $p \leq 0.05$.

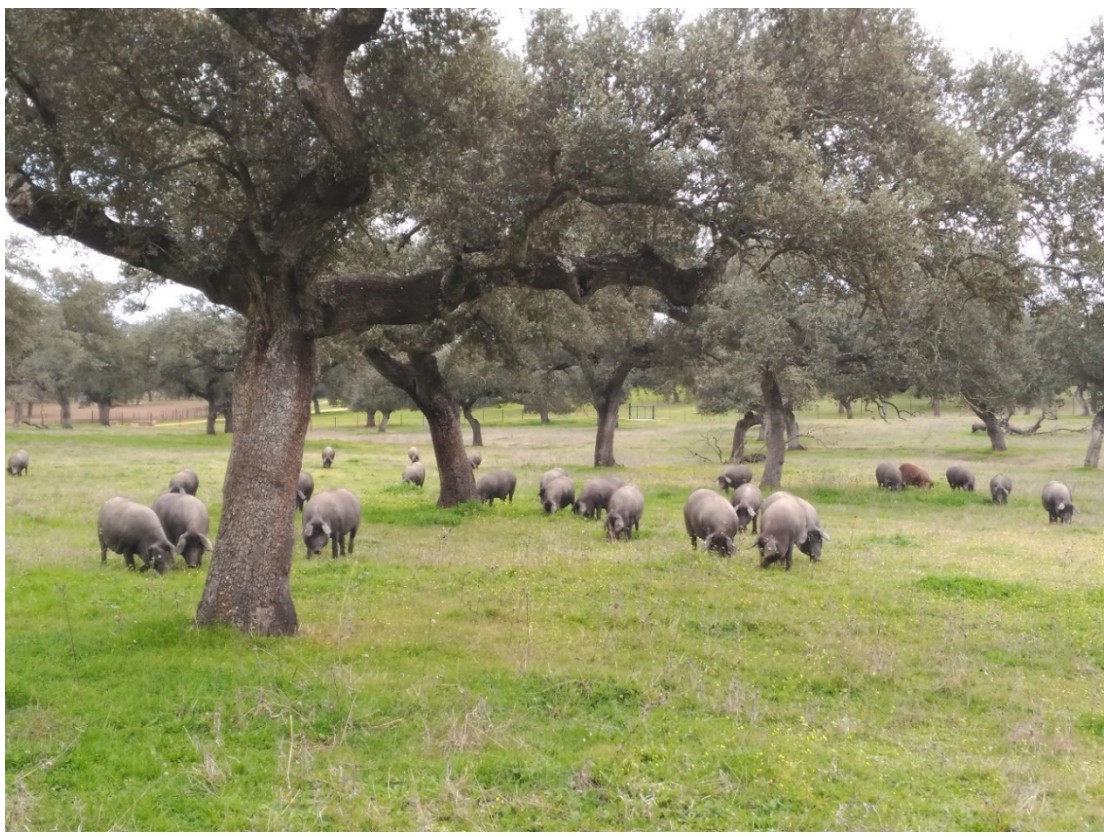

**Figure 1.** Iberian pigs in "montanera".

## 2.2. Carcass and Fatty Acid Traits

Upon reaching their allocated slaughtering age (after 30, 60 and 90 days of fattening), the pigs were fasted for 12 h, weighed immediately and transported to a commercial slaughterhouse located 100 km from the experimental farm. They were then allowed a 14 h rest period, with full access to water but no feed. The fasting period lasted 26 h, which is within the range used by the industry for Iberian pigs. At slaughter, the animals were stunned electrically, killed by exsanguination, scalded and eviscerated according to European regulations for animal care, following normal commercial abattoir procedures. The hot carcasses (including the head and loin) were weighed within 1 h post-mortem. At the packing plant, the backfat depth (total and outer, middle and inner layers) and loin area at the 14th rib level were outlined on transparent acetate paper on the left side of each carcass. The outlined loin area was measured using an LI-3100C area meter (Li-Cor Corporation, Lincoln, NE, USA) with a lens adjustable to a resolution of 1 mm$^2$. The samples were placed on the lower transparent belt of the device and allowed to pass through the LI-3100C scanning head. The resultant area was shown on the LI-3100C display. The backfat thickness was measured with a 0.5 mm precision flexible ruler (Mitutoyo, Andover, Hampshire, UK).

A loin slice at the 14th rib level (including skin, backfat and longissimus dorsi muscle) from the left half of each carcass was removed and vacuum packed in nylon/polyethylene vacuum bags, transported to the laboratory and kept frozen at −20 °C until further analyses. Before analysis, the vacuum-packed samples were thawed at 2 ± 1 °C for 24 h. In the laboratory, the backfat sample of each animal was divided into its three layers (outer, middle and inner) to analyze the fatty acid composition by the official method used in Iberian pork quality standards (ISO 5508:1990). The process was as follows: a homogeneous sample from each layer and each animal was weighed and homogenized with 50 mL of diethyl ester (2:1 *v/v*) using a Stomacher (80 or 400 depending on sample size) for 2 min. The resultant solution was filtered to separate the extracted fat (total lipid content with the solvent)

and the solid residue (connective tissue). To separate and recover the solvent, the extracted fat was put through a Rotavapor hot bath, between 40–50 °C, and then refrigerated for 5 min. To dissolve the lipids, 0.20 g of the total extracted lipids without ester was added to 4 mL n-hexane with alkenes and was shaken softly. Then, 0.2 mL of KOH (2 M methanol) was added and left to stand for 30 min. The mixture was then centrifuged at 2000 rpm for 30 s. The fatty acids were determined in 1.0 mL of the mixture by gas chromatographic analysis. The results were expressed as g/100 g of fatty acid methyl esters identified. A 2.5 cm slice was removed from the longissimus dorsi muscle at the 14th rib level for IMF content and FA analysis. The IMF was quantified according to the method described by Folch et al. [17], and the FA composition of the IMF was analyzed by the same method as the backfat layer FA determination.

*2.3. Statistical Analysis*

All the statistical analyses were performed using SAS 9.1 (SAS Institute Inc., Cary, NC, USA). The generalized linear models (GLM) and the correlation (CORR) procedures were used to analyze the data. To compare carcass traits (weight and area of the loin, intramuscular fat, and backfat thickness) in the three fattening times, the model included only the time of fattening effect. To compare the fatty acid (FA) profiles, two models were used, one was used to compare FA profile in subcutaneous and intramuscular fat in the different fattening times and the other compares the FA profiles for the three subcutaneous fat layers (inner, medium and outer) in the different fattening times as well. Both models included time of fattening, location and the interaction (time of fattening * location) effects. For all models, the least squares means were compared by Tukey's test. In addition, correlations between the FA contents in intramuscular and in subcutaneous fat were investigated by pooling the data for all fattening times.

## 3. Results and Discussion

*3.1. Growth and Carcass Traits*

The body weight (BW) at slaughter and carcass traits showed the highest mean values in the $M_{90}$ group, but this increase was not significant ($p > 0.05$) for the weight and area of the loin (Table 1). The dressing-out (carcass weight * 100/body weight at slaughter) increased significantly ($p < 0.05$) with the duration of the fattening period (75.9% and 85.6% at 30 and 90 days on "montanera," respectively), mainly due to increased fat deposition, as it was shown by the significant increase in both IMF content (229%) and subcutaneous fat thickness (170%) between $M_{30}$ and $M_{90}$ groups, while the weight and area of the loin remained practically unchanged as carcass weight increased with the fattening period length.

The average thickness of the outer, middle and inner subcutaneous adipose layers of the backfat were similar to those reported by Izquierdo et al. [18] in Iberian pigs slaughtered within the 135–175 kg range, but differed significantly from those obtained by Alfonso et al. [19] in Basque (autochthonous breed) and Large White (selected breed) pigs slaughtered at 86 and 126 kg, respectively. Eggert et al. [20] found that the inner layer thickness was proportionally lower in leaner genotypes. The mean thickness of the outer, middle and inner layers of the backfat in animals slaughtered at 90 days of finishing represented 129%, 157% and 302%, respectively, of the values obtained from $M_{30}$, showing that the individual layers do not grow at the same rate. In relation to subcutaneous fat thickness, the outer layer showed a significant ($p < 0.05$) decrease (22.83% vs. 17.13%), the middle layer showed no significant ($p > 0.05$) change, and the inner layer increased significatively ($p < 0.001$) between the $M_{30}$ and $M_{90}$ groups (32.73% vs. 42.43%). This evolution is similar to that observed when fat thickness values were related to carcass weight in each batch (1.05, 0.98 and 0.84 cm/100 kg carcass weight for the outer layer, 2.10, 1.99 and 2.01 cm/100 kg carcass weight for the middle layer, and 1.54, 1.72 and 2.11 cm/100 kg carcass weight for the inner layer), although the differences were only significant ($p < 0.05$) for the inner layer. These results contrast with those obtained by Izquierdo et al. [18] in the Iberian breed, where the only significant increase was observed in the middle layer. During finishing,

the middle and inner subcutaneous adipose layers change in their relative contribution to total backfat thickness, and the middle layer shows the greatest increase in thickness per BW unit [21]. The change in thickness of the individual layers per BW unit ranked the middle layer first (0.0040 cm/kg), followed by the outer layer (0.0031 cm/kg) and the inner layer (0.0020 cm/kg) [21]. The analysis of the ratios of individual backfat layers (outer, OBF; middle, MBF; and inner, IBF) to total backfat and for MBF:OBF, IBF:OBF and IBF:MBF in Duroc swine showed that the rate of growth of the outer layer of backfat decreased as BW increased, that the depth of the middle and inner backfat layers increased at a faster rate than the outer layer in the weight range observed, and that the middle and inner layers of backfat grew at approximately the same rate with respect to each other [22]. These contradictory results may be explained by the different diet and genotype used in those studies [19,20], and/or a difference in mean backfat level [22], weight range and sites of measurements [13]. Concerning the genotype effect, in a comparative study of the relative growth of subcutaneous fat layers in Duroc, Pietrain and Large White-sired genotypes, it was found that the outer layer appears to grow linearly for most pigs and the middle layer grows at a greater rate, while the inner backfat layer growth approaches a plateau for Pietrain-sired barrows and Large White-sired gilts, and all other pigs deposited an inner layer of backfat at an accelerating rate [20].

### 3.2. Fatty Acid Profile of Intramuscular and Subcutaneous Fat

The fatty acid composition of the fat depots was, as expected, largely a reflection of the fatty acid pattern of the dietary fat. In the present study, the most abundant fatty acids were oleic (52.94%) and palmitic acids (21.54%), followed by stearic (9.73%) and linolenic acids (8.76%) (Table 2). Likewise, a high level of MUFA (57.68%) was detected, which is the most relevant quality feature in meat from Iberian pigs [23].

Significant ($p < 0.05$) differences in majority FA between intramuscular and subcutaneous adipose tissues were observed. The anatomical variation in the fatty acid composition in pigs [16,24] could be attributed to the fact that each adipose tissue shows specific development and metabolism [25]. Comparing with the subcutaneous fat, the intramuscular fat exhibited a higher ($p < 0.05$) percentage of lauric, miristic, palmitic and palmitoleic acids, SFA, MUFA and SFA/MUFA+PUFA ratio, and a lower percentage of margaric, margaroleic, gadoleic, linoleic and linolenic acids, PUFA and unsaturation index. No significant effects ($p > 0.05$) of the adipose depot on the levels of stearic, araquic and oleic acids were observed. The percentage of palmitoleic acid in IMF was about twice of that of backfat (4.63 vs. 1.96, respectively). In contrast, the content of the gadoleic fatty acid was twice as high in backfat than in IMF (1.54 vs. 0.85, respectively). This could relate to the fact that the enzymatic activity for the formation of gadoleic acid is more active in backfat than in intramuscular fat [26].

The fatty acid profile, which is highly influenced by the characteristics of the feed [10,13,15,16,27–29], changes during fattening, as both the animal's fat deposition and the rate of fatty acid synthesis increase [30]. In the present study, the length of the fattening period significantly ($p < 0.05$) influenced the percentage content for some fatty acids, and the deposition percentage for each fatty acid differed between IMF and backfat (Table 2). No effect of the number of days of fattening was observed on lauric, myristic, palmitic, palmitoleic or gadoleic acids and MUFA for both adipose tissues. Moreover, the percentages of margaric, margaroleic, linoleic and linolenic fatty acids, PUFA and the unsaturation index decreased or increased during the fattening phase in both adipose tissues. Regarding linoleic and linolenic acids, their concentration significantly decreased ($p < 0.05$) in the two tissue depots, especially between $M_{30}$ and $M_{60}$. In the $M_{90}$ group, the percentage of both fatty acids in backfat was twice of that of intramuscular fat (9.89% vs. 4.93% and 0.78% vs. 0.33%, respectively). These results on linoleic acid diverge from those obtained from Pascual et al. [29], who indicated that the higher linoleic increase is compensated by a significantly higher decrease in palmitic and stearic acid levels. However, in our study, the opposite occurred.

Regarding the specific FAs of IMF, significant ($p < 0.05$) increases in lauric and miristic acid percentages were recorded, while the percentages of margaric, margaroleic, linoleic and linolenic acids, and PUFA showed a significant ($p < 0.05$) decrease with the increasing time of fattening.

For the subcutaneous backfat, the $M_{90}$ pigs exhibited a higher percentage of oleic and gadoleic acids and MUFA, and a lower percentage of margaric, palmitoleic, margaroleic, linoleic and linolenic acids, PUFA and unsaturation index. The percentage of the main MUFA, oleic acid, showed an increase in backfat during the days of fattening only in the two first groups (51.40% and 53.52% for $M_{30}$ and $M_{60}$, respectively). The increase in oleic acid in backfat can be influenced by the age when the animals begin the basic fattening phase on acorns and grass [8,29] and the significant differences obtained could be due to the number of days in the fattening phase, the richness in oleic acid of the acorns and the pigs' low endogenous synthesis. Additionally, our results on oleic acid agreed with those obtained by Daza et al. [9], Daza et al. [10] and Tejerina et al. [13,14]. The other backfat MUFAs, palmitoleic and margaroleic, decreased from $M_{30}$ to $M_{60}$, but the case of gadoleic acid was different as it increased in the first interval.

The interaction of both effects, fat tissue and duration of fattening phase, was significant ($p < 0.05$) only for linolenic and arachidonic fatty acids.

In Duroc and white pig breeds [31–33], the proportion of SFA and MUFA in longissimus dorsi muscle and backfat tended to rise, while the PUFA concentration and PUFA/SFA ratio fell significantly, and these changes resulted from the increasing role of de novo tissue synthesis of SFA and MUFA and the declining role of the direct incorporation of linoleic acid from the diet [13].

It was found that for Iberian pigs under intensive system and slaughtered at 50 and 115 kg, there was a decrease in SFA content and an increase in MUFA content as body weight increased, while the PUFA content was practically unchanged [27]. Similarly [29], in the subcutaneous fat of extensively fed Iberian pigs, as the fattening period increased, there was an increase in oleic acid and a decrease in stearic and palmitic acids. A similar phenomenon was found by Cava [34] for Iberian pig IMF. According to the literature, the decline in PUFA with the number of fattening days may be due to the low levels of these fatty acids in the acorns and the high fat deposition of the pigs, and the reduction in linoleic acid in the diet is compensated by the rise in palmitic and stearic acids [35]. The addition of grass to the diet resulted in high levels of linoleic and linolenic acids and a decrease in the level of palmitic acid in the intramuscular fat [36]. Additionally, as the pigs' body weight increased, the proportion of stearic acid in the backfat increased and that of linoleic acid fell [31], but, in the longissimus dorsi muscle, the proportion of stearic acid did not change, and that of the oleic acid decreased.

Finally, the values of the unsaturation index in backfat were lower than those obtained by Niñoles et al. [12] and its change as the days of the fattening phase increased did not show a clear trend. Our results agree with those obtained by other authors [12–15,28], but are lower than those obtained by Cava et al. [37], although the length of the fattening phase was not the same in those studies.

### 3.3. Fatty Acid Profile of the Three Layers of Subcutaneous Backfat

Table 3 shows the fatty acid composition of the three layers of backfat (outer, middle and inner). The fatty acid profile was significantly ($p < 0.05$) different for each layer, in agreement with the results of Monziols et al. [24]. It is widely recognized that the outer backfat layer is more unsaturated than the inner, partly by a probable preferential deposition of PUFA in the outer layer, but also as a result of the dilution effect caused by an increased FA deposition from a stimulated de novo lipogenesis in the inner layer [38].

**Table 2.** Effect of duration of fattening period on intramuscular1 (IMF) and subcutaneous fat (BF) fatty acid percentages in Iberian pigs (least squares means).

| Traits (g/100 g) [1] | Fat Tissue | | Duration of Fattening [2] | | | | | | RSD | *p*-Value | | |
| | IMF | BF | M$_{30}$ (N = 8) | | M$_{60}$ (N = 8) | | M$_{90}$ (N = 8) | | | Fat Tissue | Duration of Fattening | F.S. [3] |
| | | | IMF | BF | IMF | BF | IMF | BF | | | | |
| C 12:0 | 0.07 [a] | 0.05 [b] | 0.06 [a] | 0.05 [b] | 0.07 [a] | 0.05 [b] | 0.08 [a] | 0.06 [b] | 0.10 | *** | n.s. | n.s. |
| C 14:0 | 1.26 [a] | 1.09 [b] | 1.15 | 1.07 | 1.29 [a] | 1.09 [b] | 1.34 [a] | 1.12 [b] | 0.38 | *** | n.s. | n.s. |
| C 16:0 | 23.03 [a] | 20.06 [b] | 21.42 | 19.80 | 24.10 [a] | 20.35 [b] | 23.57 [a] | 20.05 [b] | 1.54 | *** | n.s. | n.s. |
| C 16:1 | 4.63 [a] | 1.96 [b] | 4.87 [a] | 2.28 [b] | 4.27 [a] | 1.76 [b] | 4.75 [a] | 1.85 [b] | 1.21 | *** | n.s. | n.s. |
| C 17:0 | 0.15 [b] | 0.30 [a] | 0.19 [b] | 0.35 [a] | 0.13 [b] | 0.28 [a] | 0.12 [b] | 0.27 [a] | 0.30 | *** | * | n.s. |
| C 17:1 | 0.23 [b] | 0.30 [a] | 0.29 [b] | 0.37 [a] | 0.19 [b] | 0.25 [a] | 0.19 [b] | 0.27 [a] | 0.28 | *** | *** | n.s. |
| C 18:0 | 9.73 | 9.74 | 8.92 | 9.23 | 10.50 | 10.19 | 9.77 | 9.78 | 1.16 | n.s. | * | n.s. |
| C 18:1 | 52.87 | 53.02 | 52.55 | 51.40 | 52.17 | 53.52 | 53.89 | 54.15 | 1.50 | n.s. | * | n.s. |
| C 18:2 | 6.62 [b] | 10.90 [a] | 9.07 [b] | 12.90 [a] | 5.87 [b] | 9.90 [a] | 4.93 [b] | 9.89 [a] | 1.68 | *** | *** | n.s. |
| C 18:3 | 0.39 [b] | 0.86 [a] | 0.49 [b] | 1.06 [a] | 0.35 [b] | 0.73 [a] | 0.33 [b] | 0.78 [a] | 0.52 | *** | * | *** |
| C 20:0 | 0.18 | 0.18 | 0.16 | 0.17 | 0.20 | 0.20 | 0.18 | 0.18 | 0.16 | n.s. | *** | n.s. |
| C 20:1 | 0.85 [b] | 1.54 [a] | 0.84 [b] | 1.33 [a] | 0.87 [b] | 1.69 [a] | 0.85 [b] | 1.61 [a] | 0.62 | *** | n.s. | *** |
| SFA | 34.41 [b] | 31.43 [a] | 31.90 | 30.67 | 36.29 [a] | 32.15 [b] | 35.06 | 31.46 | 1.65 | ** | * | n.s. |
| MUFA | 58.58 [b] | 56.83 [a] | 58.55 [a] | 55.38 [b] | 57.50 | 57.21 | 59.69 | 57.88 | 1.75 | * | n.s. | n.s. |
| PUFA | 7.01 [b] | 11.75 [a] | 9.55 [b] | 13.96 [a] | 6.22 [b] | 10.63 [a] | 5.26 [b] | 10.66 [a] | 0.64 | *** | *** | n.s. |
| SFA/MUFA+PUFA | 0.53 [a] | 0.46 [b] | 0.47 | 0.44 | 0.57 [a] | 0.47 [b] | 0.55 [a] | 0.46 [b] | 1.86 | *** | n.s. | n.s. |
| Unsaturation index | 2.16 [b] | 2.57 [a] | 2.48 | 2.73 | 1.95 [b] | 2.45 [a] | 2.05 [b] | 2.53 [a] | 0.28 | ** | ** | n.s. |

[1] SFA = Σ Saturated fatty acids; MUFA = Σ monounsaturated fatty acids; PUFA = Σ polyunsaturated fatty acids; Unsaturation index = [(MUFA * 1) + (PUFA * 2)]/SFA. [2] M$_{30}$ = Thirty days on fattening; M$_{60}$ = Sixty days on fattening; M$_{90}$ = Ninety days on fattening. [3] n.s.: not significant; * $p \leq 0.05$; ** $p \leq 0.01$; *** $p \leq 0.001$. [a,b] Values within a row with different superscript differ at $p \leq 0.05$ for tissue effect.

Regarding the outer layer, the percentage of miristic, palmitoleic, oleic and gadoleic acids, MUFA and unsaturation index were higher than in the other two layers, whilst palmitic and stearic FAs, SFA and the saturated/unsaturated ratio were lower. Whereas, the inner layer showed the lowest values of oleic, gadoleic and margaroleic fatty acids, MUFA and unsaturation index, and the highest in palmitic and stearic fatty acids and SFA. This can be attributed to the fact that the inner layer matures later and is therefore the last to be affected by the fatty acid composition of food. These results agreed with those obtained by Rey et al. [8] and Daza et al. [9,10,39]. In contrast with the results obtained by others authors [24,40], in our study, no significant ($p > 0.05$) differences between the layers in the concentration of linoleic acid were found. The PUFA accumulate more in either layer depending on the length of the fattening phase. The unsaturated index value, higher in the outer layer, was consistent with those obtained by Daza et al. [11] in Iberian pigs.

In our current study, the unsaturation degree of the pigs' fat depots followed a negative gradient from the outside inwards. The highest levels of SFA and SFA/MUFA+PUFA ratio were recorded in the inner layer. The MUFA content and unsaturation index differed between the three layers according to this gradient, with highest values in the outer layer and lowest in the inner layer [24]. The PUFA concentrations followed the same pattern. Differences in lipid metabolism could be the main causes of the preferential deposition of PUFA in the outer layer [38], as the lipid metabolism is lower in the outer layer than in the inner layer of subcutaneous adipose tissue.

According to the literature, the highest levels of SFA and of MUFA and PUFA were recorded in the inner and outer layers, respectively [8], although the causes of the preferential deposition of PUFA in the outer layer of the pigs' adipose tissues are not yet fully understood. It has been postulated that it may be related to the lower lipid metabolism in the outer layer [24]. On the contrary, the greater de novo lipogenesis activity in the inner layer may dilute the diet-related PUFAs with endogenous fatty acids [38]. Our results for PUFA did not agree with those of Daza et al. [10,11,39], who found the highest value in the inner layer.

The amount of fatty acid as the fattening phase is increased is different in each backfat layer (Table 3). Thus, palmitoleic, margaroleic, linoleic and linolenic fatty acids, PUFA and unsaturation index decreased as the number of days of fattening increased, whilst oleic and gadoleic acids and MUFA increased. These results agree with those obtained by Daza et al. [9–11,39], except for linoleic fatty acid, which was higher in the inner layer of backfat. The results from Rey et al. [8] also differed from ours in that only palmitic, stearic and linolenic fatty acids had the highest percentages in the inner layer.

The time of fattening significantly ($p < 0.05$) affected MUFA and PUFA in the three backfat layers and MUFA in outer and middle layers [10]. However, in the present study, there was no significant ($p > 0.05$) effect of the duration of fattening on SFA, except for margaric acid in all backfat layers and araquic acid in the outer and middle layers. These results could be due to the fact that the amount of SFA in the three backfat layers follows a similar trend, while the amount of unsaturated fatty acid is higher in one layer than the other two. Within our results, the tendencies of the different FA contents for the three backfat layers did not agree with the study from Daza et al. [10] for palmitic and stearic acids in the outer and inner layers, oleic acid and MUFA in the inner layer and SFA in the outer layer. Perhaps, an adaptation of adipose tissue to outdoor temperature, with the aim of maintaining the physical flow of lipids in the different adipose tissues [41], could be one of the causes of the differences in the fatty acid profile in the three layers.

The interaction of both effects, backfat layers and duration of fattening phase, was not significant ($p > 0.05$) for all traits of fatty acids.

**Table 3.** Effect of duration of fattening period on backfat fatty acid percentages in Iberian pigs (least squares means).

| Traits (g/100 g) [1] | Duration of Fattening [2] | | | | | | | | | RSD | p-Value [3] | | |
|---|---|---|---|---|---|---|---|---|---|---|---|---|---|
| | M30 (N = 8) | | | M60 (N = 8) | | | M90 (N = 8) | | | | | | |
| | Inner | Middle | Outer | Inner | Middle | Outer | Inner | Middle | Outer | | Inner | Middle | Outer |
| C 12:0 | 0.05 | 0.05 | 0.05 | 0.06 [a] | 0.05 [b] | 0.05 [b] | 0.06 | 0.05 | 0.05 | 0.09 | n.s. | n.s. | n.s. |
| C 14:0 | 1.06 | 1.06 | 1.09 | 1.11 [a] | 1.01 [b] | 1.14 [a] | 1.12 | 1.08 | 1.15 | 0.31 | n.s. | n.s. | n.s. |
| C 16:0 | 20.48 [a] | 19.87 [ab] | 19.04 [b] | 21.22 [a] | 20.17 [ab] | 19.65 [b] | 20.92 [a] | 19.96 [ab] | 19.25 [b] | 1.07 | n.s. | n.s. | n.s. |
| C 16:1 | 2.04 [b] | 2.08 [b] | 2.72 [a] | 1.68 [b] | 1.51 [b] | 2.08 [a] | 1.63 [b] | 1.69 [b] | 2.22 [a] | 0.68 | *** | *** | ** |
| C 17:0 | 0.36 | 0.36 | 0.33 | 0.27 | 0.28 | 0.27 | 0.27 | 0.27 | 0.27 | 0.23 | n.s. | n.s. | n.s. |
| C 17:1 | 0.32 [b] | 0.36 [a] | 0.43 [a] | 0.22 [b] | 0.24 [b] | 0.31 [a] | 0.23 [b] | 0.26 [b] | 0.33 [a] | 0.28 | n.s. | n.s. | n.s. |
| C 18:0 | 10.39 [a] | 9.74 [a] | 7.58 [b] | 11.05 [a] | 11.12 [a] | 8.40 [b] | 11.17 [a] | 10.29 [a] | 7.88 [b] | 1.30 | ** | ** | * |
| C 18:1 | 50.38 [b] | 50.89 [b] | 52.94 [a] | 52.15 [b] | 52.93 [b] | 55.47 [a] | 52.62 [b] | 53.94 [b] | 55.88 [a] | 1.47 | *** | *** | *** |
| C 18:2 | 12.44 | 12.99 | 13.28 | 9.78 | 9.99 | 9.93 | 9.50 [b] | 9.86 [ab] | 10.30 [a] | 1.25 | ** | *** | *** |
| C 18:3 | 1.09 | 1.09 | 1.00 | 0.79 [a] | 0.76 [a] | 0.65 [b] | 0.78 | 0.79 | 0.76 | 0.43 | ** | *** | *** |
| C 20:0 | 0.17 | 0.18 | 0.15 | 0.20 [ab] | 0.22 [a] | 0.18 [b] | 0.19 | 0.20 | 0.17 | 0.18 | *** | *** | *** |
| C 20:1 | 1.22 [b] | 1.34 [ab] | 1.42 [a] | 1.48 [b] | 1.71 [ab] | 1.87 [a] | 1.51 [ab] | 1.60 [b] | 1.73 [ab] | 0.49 | *** | *** | *** |
| SFA | 32.52 [a] | 31.25 [a] | 28.24 [b] | 33.91 [a] | 32.86 [a] | 29.69 [b] | 33.73 [a] | 31.86 [a] | 28.79 [b] | 1.58 | n.s. | n.s. | n.s. |
| MUFA | 53.96 [b] | 54.67 [b] | 57.51 [a] | 55.53 [b] | 56.38 [b] | 59.73 [a] | 55.99 [b] | 57.50 [b] | 60.15 [a] | 1.31 | * | ** | *** |
| PUFA | 13.53 | 14.08 | 14.28 | 10.56 | 10.75 | 10.58 | 10.28 | 10.64 | 11.06 | 0.57 | *** | *** | *** |
| SFA/MUFA+PUFA | 0.48 [a] | 0.45 [a] | 0.40 [b] | 0.51 [a] | 0.49 [a] | 0.42 [b] | 0.51 [a] | 0.47 [a] | 0.41 [b] | 1.61 | n.s. | n.s. | n.s. |
| Unsaturation index | 2.50 [b] | 2.67 [b] | 3.06 [a] | 2.27 [b] | 2.38 [b] | 2.73 [a] | 2.28 [b] | 2.48 [b] | 2.86 [a] | 0.23 | n.s. | n.s. | n.s. |

[1] SFA = Σ saturated fatty acids; MUFA = Σ monounsaturated fatty acids; PUFA = Σ polyunsaturated fatty acids; Unsaturation index = [(MUFA × 1) + (PUFA × 2)]/SFA. [2] M30 = Thirty days on fattening; M60 = Sixty days on fattening; M90 = Ninety days on fattening. [3] n.s.: not significant; * $p \leq 0.05$; ** $p \leq 0.01$; *** $p \leq 0.001$. [a,b] Values within a row with different superscript differ at $p \leq 0.05$.

### 3.4. Relationship between Intramuscular and Backfat Fatty Acids

The fatty acid composition of the intramuscular fat largely determines the quality of the pork, and the subcutaneous fat is usually used to characterize and determine the commercial grade of Iberian meat according to the production system (management and feeding) [42]. Based on these two facts, several studies have focused on the relationship between the fatty acid profile of intramuscular and subcutaneous fat in heavy pigs [43]; however, no research has studied this relationship in Iberian pigs. As indicated by Niñoles et al. [12], if a significant relationship between the composition of both types of fat could be found, then a fast, objective, cost-efficient analysis and some non-destructive techniques could be developed to determine the quality of pork [44]. The simple correlation coefficients between the same fatty acids from intramuscular and subcutaneous depots and their layers in Iberian pigs fattened on a "montanera" production system are shown in Table 4. The results show a strong relationship between the fatty acid profiles of both tissues, except for lauric and myristic acids. There was a strong relationship between the SFA and MUFA, which is in agreement with the study from Yang et al. [43]. However, the PUFAs showed no significant ($p > 0.05$) relationship between tissues, which differs from the findings of Yang et al. [43]. The quality-wise most important fatty acid in Iberian pig, the oleic acid, showed a significant correlation coefficient ($p < 0.05$) between fatty depots. This could be due to its constant accumulation in both adipose tissues. In lean pigs, as found in Landrace and Duroc, the high degree of association between the fatty acids suggests that as intramuscular fat content increases, there is a rapid dilution of polyunsaturated fatty acids by saturated and monounsaturated fatty acids [45]. Our results, as stated by Yang et al. [43], indicate that the fatty acid depositions of both adipose depots are associated with each other to some extent.

**Table 4.** Pearson's correlation coefficients between fatty acids of intramuscular fat and subcutaneous fat and their layers in Iberian pigs.

|            | Inner      | Medium     | Outer      | BF         |
|------------|------------|------------|------------|------------|
| IMF_C12_0  | 0.27 ns    | 0.17 ns    | 0.40 *     | 0.05 ns    |
| IMF_C14_0  | 0.57 **    | 0.32 ns    | 0.52 *     | 0.49 *     |
| IMF_C16_0  | 0.64 **    | 0.60 **    | 0.66 **    | 0.65 **    |
| IMF_C16_1  | 0.74 ***   | 0.74 ***   | 0.76 ***   | 0.76 ***   |
| IMF_C17_0  | 0.91 ***   | 0.89 ***   | 0.86 ***   | 0.90 ***   |
| IMF_C17_1  | 0.92 ***   | 0.92 ***   | 0.91 ***   | 0.92 ***   |
| IMF_C18_0  | 0.87 ***   | 0.91 ***   | 0.83 ***   | 0.90 ***   |
| IMF_C18_1  | 0.72 ***   | 0.71 ***   | 0.59 **    | 0.69 ***   |
| IMF_C18_2  | 0.89 ***   | 0.86 ***   | 0.79 ***   | 0.86 ***   |
| IMF_C18_3  | 0.79 ***   | 0.78 ***   | 0.78 ***   | 0.80 ***   |
| IMF_C20_0  | 0.81 ***   | 0.78 ***   | 0.72 ***   | 0.82 ***   |
| IMF_C20_1  | 0.70 ***   | 0.62 **    | 0.53 *     | 0.63 **    |

n.s.: not significant; * $p \leq 0.05$; **: $p \leq 0.01$; *** $p \leq 0.001$.

## 4. Conclusions

The backfat of Iberian pigs fed with acorn is more unsaturated than the intramuscular backfat. The duration of the "montanera" fattening system affected the content of fatty acids. The change in oleic acid was higher in backfat than in intramuscular fat, although the values in both adipose tissues were quite similar. The strong relationship between the fatty acid profiles of adipose tissues might be used to predict intramuscular fat profile from backfat samples since the latter are cheaper to collect.

**Author Contributions:** Conceptualization and methodology, all authors; Formal analysis, software, data curation, data processing, A.G., F.P. and M.I.; Statistical analysis, A.G. and F.P.; Validation and investigation, A.G., F.P. and D.A.; Supervision, project administration, M.I. and F.I.H.-G.; Data acquisition, D.A. and M.I.; All authors were involved in developing, writing, commenting, editing and reviewing the manuscript. All authors have read and agreed to the published version of the manuscript.

**Funding:** This research was funded by INIA RTA 2007-000-93-00-00 and by FEDER.

**Conflicts of Interest:** The authors declare no conflict of interest.

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
