# Peer review of "Effect of Fattening Period Length on Intramuscular and Subcutaneous Fatty Acid Profiles in Iberian Pigs Finished in the Montanera Sustainable System"

_sustainability, doi:10.3390/su12197937_

Round 1

Reviewer 1 Report

The author explored the fatty acid profiles in intramuscular (IMF) and subcutaneous fat (BF) of Iberian pigs with the change of fattening period length. They found that the fatty acid profiles in IMF and BF were influenced by the duration of fattening period. Each type of fatty acids (e.g. saturated, mono-unsaturated, or poly-unsaturated fatty acids) has a different tendency of accumulation with fattening duration, depending on the location of adipose tissues in BF. The authors found the strong relationship between fatty acid profiles of IMF and BF, and suggested that this relationship could be used to predict one profile from another and be available for sampling or analytical reasons. This manuscript would be interesting enough to be published in this journal. However, the following should be fixed or added in the manuscript.

Comments

  1. In line 90 of page 3, the author mentioned that fatty acids from samples were analyzed with the official method used in Iberian pork quality standards. It would be better to include the reference which shows the details of analytical method, such as chromatographic column, temperature gradient, and etc.
  2. In line 92 of page 3, ‘ester diethyl’ is supposed to be ‘diethyl ester’.
  3. In line 99 of page 3, ‘chromatographic gas analysis’ is supposed to be ‘gas chromatographic analysis’.
  4. Superscript on each of the mean value in tables was so confusing. It is not clear which groups are significantly different from a group.
  5. RSD in each table is also confusing. Was the RSD calculated from each group in the table or from all groups in total?
  6. Which duration of fattening is used for the Pearson correlation coefficients in table 4? The author stated that the Pearson correlation coefficients were increased with the duration of the fattening period. The details should be included as a table, because this could be useful for potential applications.

Author Response

We appreciate the effort of reviewer. We believe that it has served to substantially improve the manuscript. We have made the corrections you have suggested. We have also tried to improve the Language (more simple and clear sentences). The other efforts have focused on clarifying the methodology used.The authors.

Response

1/ In line 90 of page 3, the author mentioned that fatty acids from samples were analyzed with the official method used in Iberian pork quality standards. It would be better to include the reference which shows the details of analytical method, such as chromatographic column, temperature gradient, and etc.

Answer: We have added the reference of methodology official to analyses fatty acid composition in Iberian pig

2/ In line 92 of page 3, ‘ester diethyl’ is supposed to be ‘diethyl ester’.

Answer: We have changed ester diethyl’ by ‘diethyl ester’.

3/ In line 99 of page 3, ‘chromatographic gas analysis’ is supposed to be ‘gas chromatographic analysis’.

Answer: We have changed chromatographic gas analysis’ by ‘gas chromatographic analysis’.

4/ Superscript on each of the mean value in tables was so confusing. It is not clear which groups are significantly different from a group.

Answer: We have reformulate the information of Table 2 and 3 following the comments of other review and for improve the understanding of each superscript

5/ RSD in each table is also confusing. Was the RSD calculated from each group in the table or from all groups in total?

Answer: The RSD was calculated from all groups in total because we think that this valor gives more information of the variance in the all data collection than the SD of each group. But if you think that the SD for each group is better, we will change it in this sense.

6/ Which duration of fattening is used for the Pearson correlation coefficients in table 4? The author stated that the Pearson correlation coefficients were increased with the duration of the fattening period. The details should be included as a table, because this could be useful for potential applications.

Answer: We have removed the following part of the sentence because we had not evaluate the correlation between fatty acids profile as increase the fattening phase: “, and the value and statistical significance of the coefficient of correlation in our study increased with the duration of the fattening period.”

Reviewer 2 Report

This paper describes in iberian pig breed the effects of fattening duration on animal performance and fatty acid composition in different fat compartments.

It provides new insights in the domain and is thus valuable to be published.

The main concerns are related to the statistical model used that seems unappropriated. More details are provided in the joined file.

L64: the very high level of protein in pasture grass should be verified.
2.3. Statistical model: time and weight are highly correlated. What are the impacts on the results? Nesting is not clear. In my opinion, "Location" is fixed and Animal nested in time. See statistician. May be a mixed model with covariance structure would be better. Use of carcass weight as covariate change largely the lsmeans. Is this covariate required? L296: interaction could not be measured with the proposed model.
4. Conclusions: like a summary. Need to be rewritten.  

Author Response

We appreciate the effort of reviewer. We believe that it has served to substantially improve the manuscript. We have made the corrections you have suggested. We have also tried to improve the Language (more simple and clear sentences). The other efforts have focused on clarifying the methodology used.

The authors.

Response

1/ L64: the very high level of protein in pasture grass should be verified.

Answer: We have verified the level of protein in the pasture grass that appear in our work. The data is correct. We choose this reference because the experience was made in the same research farm although we know that the level of protein depends on the time of year and soil type.

2/ 2.3. Statistical model: time and weight are highly correlated. What are the impacts on the results? Nesting is not clear. In my opinion, "Location" is fixed and Animal nested in time. See statistician. May be a mixed model with covariance structure would be better. Use of carcass weight as covariate change largely the lsmeans. Is this covariate required?

Answer: We have rewritten the statistical part for improve the understanding of the all analyses done.

3/ L296: interaction could not be measured with the proposed model.
Answer: We have reformulated the information of the tables 2 and 3 for improve the understanding. In this sense the interaction in the improve statistical model will can be measured

4/ Conclusions: like a summary. Need to be rewritten.  Principio del formulario

Answer: We have rewritten the conclusions section following your indications

Round 2

Reviewer 2 Report

I agree with the revision.

Just, the Conclusions look like a presentation of results. As far as I know, a conclusion does not refer to group name, probabilities or other consideration already present in the Result section. A conclusion is more as a popularized overview of the results (a "take-home message").  

Line 76: remove a comma and replace to a point.

Statistical analysis: I am not sure that "Location" effet could be considered as a simple fixed effect (because different samples stemmed from a same animal). But OK.

In my opinion, there are a lot of excessive decimals in the data.

Author Response

We appreciate the effort of reviewer. We believe that it has served to substantially improve the manuscript. We have made the corrections you have suggested.

The authors.

1/ Just, the Conclusions look like a presentation of results. As far as I know, a conclusion does not refer to group name, probabilities or other consideration already present in the Result section. A conclusion is more as a popularized overview of the results (a "take-home message").  

Answer: We have rewritten the conclusions following your considerations

2/ Line 76: remove a comma and replace to a point.

Answer: The phrase of line 76 is: “The fasting period lasted 26 h, which is within the range used by the industry for Iberian pigs.” We do not know if there is a error in the line which you say because if we change coma by point the phrase have not sense but if you think that is better make the change we will do it.

3/ Statistical analysis: I am not sure that "Location" effet could be considered as a simple fixed effect (because different samples stemmed from a same animal). But OK.

Answer: Putting the effect of the layer as an effect to be evaluated within the GLM model is a tender way to evaluate the different layers of fat tissue, as we have seen in some articles in the bibliography.

4/ In my opinion, there are a lot of excessive decimals in the data.

Answer: We have followed your considerations changing decimals of p_value by one, two o three * or n.s.. Also we reduce the number of decimals to 2 in RSD value